# Sialic Acid-Loaded Nanoliposomes with Enhanced Stability and Transdermal Delivery for Synergistic Anti-Aging, Skin Brightening, and Barrier Repair

**DOI:** 10.3390/pharmaceutics17080956

**Published:** 2025-07-24

**Authors:** Fan Yang, Hua Wang, Dan Luo, Jun Deng, Yawen Hu, Zhi Liu, Wei Liu

**Affiliations:** 1College of Life Science and Technology, Huazhong University of Science and Technology, Wuhan 430074, China; yangfanfenix@outlook.com; 2Research & Development Center, Mageline Biology Tech Co., Ltd., Wuhan 430000, China; wanghua@mageline.cn (H.W.); huyawen@mageline.cn (Y.H.); 3Wuhan Bestcarrier Biotechnology Ltd., Wuhan 430075, China; laurel565118@163.com (D.L.); 15277662826@163.com (J.D.); 4National Engineering Research Center for Nanomedicine, Huazhong University of Science and Technology, Wuhan 430075, China

**Keywords:** sialic acid, nanoliposomes, transdermal delivery, anti-aging, barrier-repair, skin-brightening

## Abstract

**Objectives**: Sialic acid (SA), a naturally occurring compound abundantly found in birds’ nests, holds immense promise for skincare applications owing to its remarkable biological properties. However, its low bioavailability, poor stability, and limited skin permeability have constrained its widespread application. **Methods**: To overcome these challenges, SA was encapsulated within nanoliposomes (NLPs) by the high-pressure homogenization technique to develop an advanced and efficient transdermal drug delivery system. The skincare capabilities of this novel system were comprehensively evaluated across multiple experimental platforms, including in vitro cell assays, 3D skin models, in vivo zebrafish studies, and clinical human trials. **Results**: The SA-loaded NLPs (SA-NLPs) substantially improved the transdermal penetration and retention of SA, facilitating enhanced cellular uptake and cell proliferation. Compared to free SA, SA-NLPs demonstrated a 246.98% increase in skin retention and 1.8-fold greater cellular uptake in HDF cells. Moreover, SA-NLPs protected cells from oxidative stress-induced damage, stimulated collagen synthesis, and effectively suppressed the secretion of matrix metalloproteinases, tyrosinase activity, and melanin production. Additionally, zebrafish-based assays provided in vivo evidence of the skincare efficacy of SA-NLPs. Notably, clinical evaluations demonstrated that a 56-day application of the SA-NLPs-containing cream resulted in a 4.20% increase in L*, 7.87% decrease in b*, 8.45% decrease in TEWL, and 4.01% reduction in wrinkle length, indicating its superior brightening, barrier-repair, and anti-aging effects. **Conclusions**: This multi-level, systematic investigation strongly suggests that SA-NLPs represent a highly promising transdermal delivery strategy, capable of significantly enhancing the anti-aging, barrier-repair, and skin-brightening properties of SA, thus opening new avenues for its application in the fields of dermatology and cosmeceuticals.

## 1. Introduction

Sialic acid (SA), a naturally occurring monosaccharide predominantly extracted from edible birds’ nests (EBNs), constitutes approximately 10% of EBN’s bioactive components. As a terminal modifier of glycan chains in mucopolysaccharides, glycoproteins, and glycolipids, SA primarily exists as N- or O-glycosides, serving as a critical structural determinant for the functional diversity of carbohydrate complexes. Beyond its roles in intercellular recognition and immune regulation, SA modulates neural signaling, pathogen-host interactions, and synaptic plasticity, while maintaining redox homeostasis through suppression of inflammatory pathways and enhancement of antioxidant enzyme activity [1,2]. These multifaceted biological properties have positioned SA as a promising candidate for dermatological applications, particularly in anti-aging and skin-brightening formulations.

In recent years, SA has garnered significant attention for its dual efficacy in melanogenesis inhibition and collagen synthesis promotion. Mechanistically, SA directly inhibits tyrosinase—a copper-containing enzyme central to melanin biosynthesis—by competitively binding to its active site, thereby blocking both the monophenolase (hydroxylation of l-tyrosine to l-DOPA) and diphenolase (oxidation of l-DOPA to dopaquinone) activities. In vitro studies demonstrate that SA reduced the melanin production in B16 mouse melanoma cells and A375 human melanoma cells. Furthermore, in the 3D human skin model constructed by keratinocytes and melanocytes, SA penetrated the keratinocyte layer, effectively reducing melanocyte density and melanin content [3,4].

Beyond its skin-lightening effects, SA exhibits robust anti-aging potential. In human dermal fibroblasts, SA stimulates collagen synthesis by activating the TGF-β/Smad signaling pathway. Concurrently, SA suppresses matrix metalloproteinase-1 (MMP-1) and MMP-3 expression by inhibiting AP-1 and NF-κB transcriptional activity, thereby mitigating UV-induced collagen degradation. Additionally, SA exhibits anti-inflammatory, anti-oxidation, and wound-healing activities in human skin keratinocytes and fibroblasts. SA enhances skin barrier function by upregulating filaggrin and its associated genes expression in keratinocytes, increasing stratum corneum hydration and reducing transepidermal water loss (TEWL). In terms of antioxidant effectiveness, SA exhibits remarkable free radical scavenging capabilities, enhances antioxidant enzyme systems, and inhibits oxidative stress-related signaling pathways. Studies have confirmed that the application of sialic acid can significantly mitigate UV-induced oxidative stress-mediated cellular damage in both HaCaT cells and epitheliums [5,6].

Despite these promising benefits, the application of SA is hindered by its inherent limitations, including poor skin permeability, rapid degradation, and low bioavailability in conventional formulations [7,8]. Owing to its strong hydrophilic nature, only a small amount of SA can traverse the lipophilic stratum corneum via passive diffusion to reach the living epidermis, with very little of it accumulating in the dermis. Additionally, SA is prone to enzymatic hydrolysis and oxidation, severely restricting its effectiveness in skincare applications. These limitations underscore the need for advanced delivery systems to enhance SA’s stability, permeation, and targeted delivery [9,10]. In recent years, a variety of nanocarrier systems beyond liposomes have been developed to enhance the dermal delivery of cosmeceutical agents, particularly aiming to improve skin penetration, stability, and bioavailability. For instance, Reis-Mansur et al. developed oil-in-water nanoemulsions incorporating buriti oil and spray-dried Aloe vera extract, alongside conventional chemical UV filters. The optimized formulation exhibited favorable physicochemical properties, broad-spectrum UV protection, and good skin compatibility [11]. Similarly, Wang et al. reported a water-soluble fullerene nanocomposite encapsulated in sodium hyaluronate, which showed superior anti-inflammatory activity in a rat ear swelling model compared to conventional cosmetic agents, highlighting its potential in skincare and dermatological applications [12]. These findings underscore the advantages of advanced nanocarrier platforms for topical cosmeceutical delivery.

Nevertheless, among these various systems, nanoliposomes (NLPs) remain a highly versatile and biocompatible choice, especially suitable for encapsulating sensitive compounds such as SA [13]. Key advantages of NLPs include the ability to encapsulate both hydrophilic and lipophilic substances, improved physicochemical stability, enhanced skin penetration, and controlled release properties [14]. The unique bilayer structure of NLPs enables its capability to encapsulate lipophilic ingredients in the lipid membrane and incorporate hydrophilic compounds in the aqueous core [15,16]. By lipid membrane encapsulation, NLPs could protect the encapsulated active ingredient from degradation and achieve sustained release to ensure prolonged therapeutic effects. Furthermore, NLPs could enhance skin permeation, possibly due to their small particle size, high affinity between the lipid bilayer and the skin lipids, a hydration effect by forming an occlusive film on the skin surface, fusion with cellular membranes or lipid bilayers in the stratum corneum, and entry through follicular and glandular pathways. Through multifaceted mechanisms, the skin permeation of active ingredients can be considerably enhanced via NLP-mediated delivery [17,18]. Furthermore, the composition and properties of NLPs can be optimized to suit specific drug characteristics, making them versatile tools in transdermal and topical drug delivery [19]. For instance, NLPs encapsulating botulinum toxin A achieved nearly an eightfold increase in dermal penetration compared to free BTXA [20]. Similarly, palmitoylethanolamide-loaded NLPs promoted efficient transdermal delivery and demonstrated multiple skin benefits, including anti-inflammatory effects and skin barrier enhancement, highlighting its potential for topical applications [21]. In our group, a variety of NLP-based delivery systems have been successfully developed to enhance the transdermal delivery efficiency of a variety of active ingredients; e.g., resveratrol was encapsulated within NLPs, demonstrating significant enhancement in both transdermal permeability and cutaneous retention efficacy of the encapsulated compound, while concurrently improving cellular internalization. Furthermore, comparative analysis revealed that the nanoencapsulated formulation exhibited superior cosmeceutical performance relative to free resveratrol [22]. In addition, compared with traditional formulation, the developed phenylethyl resorcinol–cationic NLPs not only showed significantly higher efficiency in suppressing melanin synthesis but also had much better skin penetration ability, thereby significantly enhancing its skin-whitening performance. These successes validate NLPs’ potential for SA delivery.

Despite SA’s promising potential in anti-aging and skin whitening functionality, its application still remains limited due to insufficient mechanistic and formulation studies. The current literature is deficient in comprehensive, systematic evaluations of the skincare efficacy of SA-encapsulated NLPs across diverse relevant experimental models, especially in 3D skin models and in vivo zebrafish models. Therefore, in order to enhance the skincare efficacy of SA, we developed SA-loaded NLPs (SA-NLPs) via high-pressure homogenization with uniform particle size distribution, high encapsulation efficiency, and efficient transdermal delivery capacity. The stability, biocompatibility, and skin permeation ability were investigated, and the skincare efficacy of SA-NLPs, including anti-aging, skin whitening, and skin barrier repair functionality, was systematically evaluated at the cellular level, in a 3D skin model, and in vivo in zebrafish. The SA-NLPs with high stability and skin permeability show promising effects in anti-aging, skin brightening, and barrier repair. SA-NLPs, engineered with enhanced stability and superior transdermal permeability, exhibit multi-functional efficacy in anti-aging, skin brightening, and barrier repair, positioning them as a transformative platform for advanced skincare formulations.

## 2. Materials and Methods

### 2.1. Materials

Sialic acid (*N*-acetylneuraminic acid; chemical name: 5-acetamido-3,5-dideoxy-D-glycero-D-galacto-non-2-ulopyranosonic acid (Appendix A); purity ≥ 99%) was purchased from Wuhan Zhongke Guanggu Green Biotechnology Co., Ltd. (Wuhan, China), and stored in a cool and dry place. Phosphatidylcholine from soybean was obtained from Shanghai Taiwei Pharmaceutical Co., Ltd. (Shanghai, China). Pentanediol was purchased from B&B Korea Co., Ltd. (Seoul, Republic of Korea). PEG-40 (CO40) was purchased from BASF Co., Ltd. (Ludwigshafen, Germany). TW-80, sodium lauryl sulfate (SLS) hydroxyacetophenone, and phenylthiourea (PTC) were purchased from Aladdin Holdings Group Co., Ltd. (Beijing, China). Transcutol CG (TCG) was purchased from Guangzhou Rainbow household chemical Co., Ltd. (Guangzhou, China). Polyglycerol-4 oleate was purchased from Sungrow Chemical Co., Ltd. (Tokyo, Japan). Dulbecco’s modified Eagle’s medium (DMEM), fetal bovine serum (FBS), phosphate-buffered saline (PBS, pH 7.4), penicillin, streptomycin, and trypsin–EDTA were obtained from Gibco (Gaithersburg, MD). Hydrogen peroxide (H_2_O_2_) was purchased from Sigma-Aldrich Co. (St. Louis, MO, USA). The cell counting kit-8 (CCK-8) assay kit was obtained from Dojindo (Kumamoto, Japan). The reactive oxygen species (ROS), matrix metalloproteinases-1 (MMP-1), MMP-3, collagen type I (COL-I), COL-IV, tyrosinase, melanin, filaggrin (FLG), aquaporin3 (AQP3), claudin-1(CLND-1), and hyaluronic acid (HA) assay kits were obtained from Beyotime (Shanghai, China). Paraformaldehyde (4%) was purchased from Biosharp (Hefei, China). H_2_DCFDA was purchased from Thermo Fisher Scientific Inc. FastPure Cell/Tissue Total RNA Isolation Kit V2, HiScript III RT SuperMix for qPCR (+gDNA wiper), PerfectStart^®^ Green qPCR SuperMix were purchased from Vazyme Biotech Co., Ltd. (Nanjing, China).

### 2.2. Preparation and Characterization of SA-NLPs

SA-NLPs were prepared using a high-pressure homogenization technique. Briefly, 3% (*w*/*w*) sialic acid, 1.4% (*w*/*w*) trometamol, 10% (*w*/*w*) glycerol, and 53.6% (*w*/*w*) water were mixed to produce phase A;1% (*w*/*w*) lecithin and 5% 1,5-pentanediol were mixed to produce phase B; and 1% (*w*/*w*) vitamin E, 5% (*w*/*w*) TCG, 10% (*w*/*w*) CO40 and 10% (*w*/*w*) TW80 were mixed to produce phase C. The Phase A solution was stirred at 45 °C until completely dissolved and transparent. Phase B was then stirred at 65 °C until homogeneous and transparent, after which it was cooled to 45 °C. Phase C was first combined with Phase B under continuous stirring at 45 °C. Subsequently, Phase A was added to the mixture and stirred uniformly at 45 °C, followed by filtration. The final formulation consisted of 68% (*w*/*w*) Phase A, 6% (*w*/*w*) Phase B, and 26% (*w*/*w*) Phase C, yielding a total weight ratio of A:B:C = 68:6:26. The mixture was then homogenized using an AMH-3 microjet high-pressure homogenizer (Antos Nanotechnology, Suzhou, China) at 800 bar for three cycles. Finally, the SA-NLPs sample was purified by centrifugation at 15,000× *g* for 30 min using an ultrafiltration tube (MWCO 30 kDa, Amicon Ultra, Millipore. Billerica, MA, USA) to remove unencapsulated SA and obtain purified SA-NLPs. RhoB-loaded NLPs (RhoB-NLPs) were prepared using the same procedure, except that RhoB was incorporated into Phase A as a replacement for the active ingredient. The prepared SA-NLPs were stored under various conditions: lighting, room temperature (RT), refrigeration (4 °C), freezing (−20 °C), and high temperature (45 °C). Particle size and PDI were assessed after 14 and 28 days of storage.

The drug loading efficiency (*DLE*) and encapsulation efficiency (*EE*) of the SA-NLPs were determined using an ultrafiltration-centrifugation method. SA content was analyzed using a BOCL 101 high-performance liquid chromatography (HPLC) system (Shimadzu Instruments, Columbia, MD, USA) equipped with a ChromCore AR C18 column (4.6 mm × 250 mm, 5.0 µm, Suzhou, China). The mobile phase for SA detection consisted of acetonitrile: 0.1% phosphoric acid water = 55:45 (*v*/*v*). Chromatographic conditions included a column temperature of 30 °C, UV detection wavelength of 215 nm, injection volume of 20 μL, and flow rate of 1 mL/min. *DLE* and *EE* were calculated using the following equations:$$DLE(\%)=\frac{W_{e}}{W_{m}}\times 100$$


$$EE(\%)=\frac{W_{e}}{W_{e}+W_{f}}\times 100$$


In which *W*_e_ denotes the mass of the active ingredients encapsulated in the nanocarrier, *W*_m_ represents the total mass of the nanocarrier, and *W*_f_ signifies the mass of the free active ingredients not encapsulated in the nanocarrier.

The particle size, polydispersity index (PDI), and zeta potential of the SA-NLPs were characterized using dynamic light scattering (DLS) using a Zetasizer/Nano-ZS90 instrument (Malvern Instruments, Malvern, UK). Prior to DLS analysis, the SA-NLPs sample was diluted 100-fold in deionized water to ensure optimal measurement quality. The morphology of the SA-NLPs was observed via transmission electron microscopy (TEM, HT7700, Hitachi, Tokyo, Japan). For TEM imaging, the SA-NLPs underwent 400-fold dilution in deionized water, deposition onto a copper grid, staining with 1% phosphomolybdic acid, and air drying before observation.

### 2.3. Cell Culture

HaCaT (American Type Culture Collection, ATCC, Manassas, VA, USA), HSF (SynthBio, Hefei, China), and B16 (SynthBio, Hefei, China) cells were cultured in DMEM supplemented with 10% FBS and 1% penicillin/streptomycin, maintained at 37 °C in a humidified 5% CO_2_ atmosphere.

### 2.4. In Vitro Cytotoxicity

In vitro cytotoxicity of the SA-NLPs was evaluated using the CCK-8 assay. A total of 100 μL HaCaT cells in the logarithmic growth phase were seeded into 96-well plates at a density of 1.5 × 10^4^ cells/well, while HDF cells were seeded at a density of 8 × 10^3^ cells/well. After 24 h incubation (37 °C, 5% CO_2_), the cells were treated with 100 μL DMEM complete medium containing SA-NLPs. Tested concentrations corresponded to encapsulated SA at 10, 20, 40, 80, and 160 μg/mL. Parallel wells received 100 μL of DMEM complete medium with free SA matching the SA concentrations. The control group received 100 μL of DMEM complete medium only. Each group had three replicates. Following an additional 24 h incubation, cell viability was quantified using the CCK-8 assay.

### 2.5. Chicken Embryo Chorionic Allantoic Membrane Eye Irritation Test (HET-CAM)

A tenfold dilution of SA-NLPs in normal saline yielded a test sample containing 10% SA-NLPs (equivalent to 0.3% SA). A total of 0.2 mL of the sample was applied to the chorioallantoic membrane (CAM) surface. Vascular changes were monitored for 5 min, and the initial time of congestion, hemorrhage, and coagulation in the CAM vessels was recorded. The irritation score (IS) was calculated as follows:IS = [(301 − secH) × 5 + (301 − secL) × 7 + (301 − secC) × 9]/300

In which secH represents the initial time of hyperemia (s), secL represents the initial time of hemorrhage (s), and secC represents the initial time of coagulation (s). IS values were used for classification: 0~0.9 (no irritation), 1.0~4.9 (mild irritation), 5.0~8.9 (moderate irritation), and 9~21.0 (severe irritation).

### 2.6. In Vitro Release Study

To investigate the in vitro release profiles of free SA and SA-NLPs, 1 mL of 10% SA-NLPs (equivalent to 0.3% SA) and 1 mL of free SA solution (at the same SA concentration) were each sealed in a dialysis bag (MWCO 14 kDa) and immersed in 80 mL of PBS containing 20% (*w*/*w*) 1,2-propanediol as the release medium. The samples were incubated in a shaking water bath at 32 °C and 120 rpm. At specific time intervals (0.5, 1, 2, 4, 6, 8, 10, and 12 h), 1 mL of the release medium was withdrawn and replenished with an equal volume of fresh medium. Subsequently, the concentration of SA released were quantified by HPLC.

### 2.7. In Vitro Skin Permeation Study

Vertical Franz diffusion cells equipped with excised porcine skin were employed for skin permeation investigation. The skin was mounted between the donor and receiver chambers. The donor compartments were loaded with either 0.5 g of SA-NLPs essence (containing 5% SA-NLPs, equivalent to 0.15% SA) or 0.5 g of free SA solution matched for SA concentration. PBS served as the receptor medium. The experiment was carried out at 32 °C with magnetic stirring. Aliquots (0.5 mL) of receptor fluid were sampled and replaced with fresh PBS at 4, 6, 8, 12, and 24 h. SA content was quantified via HPLC to determine cumulative permeation per unit area. After 24 h, the skin was removed, cleaned, minced, and homogenized. The homogenate was then centrifuged with methanol, and the supernatant was analyzed by HPLC to quantify retained SA per unit area.

Transdermal delivery behavior was further visualized using RhoB-loaded NLPs (RhoB-NLPs) with RhoB as a model payload. Porcine skin was similarly mounted in Franz cells. Donor chambers received 0.5 g of either RhoB-NLPs solution (5% RhoB-NLPs) or free RhoB solution at equivalent payload concentration. PBS was used as the receptor fluid. During the experiment, the stirring was maintained at 32 °C. After 2 or 4 h, residual material was gently wiped from the skin surface, followed by thorough washing and drying. Cryosectioned samples were examined under a fluorescence microscope (IX71, Olympus, Japan; E_x_ 495 nm, E_m_ 519 nm) to visualize RhoB distribution.

### 2.8. Cellular Uptake Study

To visualize cellular uptake, RhoB was incorporated into NLPs as a fluorescent tracer. HDF cells in the logarithmic growth phase were seeded into 35 mm confocal dishes at a density of 3.0 × 10^5^ cells per dish and cultured for 24 h. Cells were then treated with DMEM containing either free RhoB or RhoB-NLPs at equivalent RhoB concentration. After 2 or 4 h incubation, the medium was removed, and the cells were washed three times with PBS. Subsequently, the cells were stained with DAPI solution for 15 min and fixed with 4% paraformaldehyde. Observations were conducted using a laser confocal microscope with excitation and emission wavelengths of 360 and 460 nm, respectively.

For quantitative cellular uptake analysis, flow cytometry was performed. HDF cells were seeded into 6-well plates (2.0 × 10^5^ cells per well) and incubated for 24 h. Following medium removal, each well was supplemented with DMEM containing either free RhoB or RhoB-NLPs at equal RhoB concentration, with untreated cells serving as the negative control. After 2 or 4 h further incubation, the cells were washed with cold PBS, trypsinized, centrifuged, and resuspended in 0.5 mL cold PBS solution. Fluorescence intensity was analyzed using flow cytometry (cytoFLEX, Beckman Coulter, Inc., Brea, CA, USA).

### 2.9. Cell Proliferation Assay

HaCaT and HDF were seeded into 96-well plates at 8 × 10^3^ cells per well and cultured at 37 °C for 24 h. Subsequently, each well was supplemented with 100 μL of DMEM complete medium containing the SA-NLPs at SA concentrations of 20, 40, and 80 μg/mL or free SA at equivalent active ingredient concentrations. The negative control (NC) group received 100 μL DMEM only. Following a 48 h incubation, the cell proliferation rate was measured using the CCK-8 assay.

The 5-ethynyl-2′-deoxyuridine (EdU) assay was also performed to assess the cell proliferation. HaCaT and HDF cells were seeded in 15 mm glass-bottom dishes at a density of 2 × 10^5^ cells per dish, incubated for 24 h, and treated with SA-NLPs/free SA as described above. The NC group was treated with DMEM, while the positive control (PC) group received 40 μg/mL VC. After incubation for 48 h, the cells were processed according to the manufacturer’s instructions using a BeyoClick EdU Cell Proliferation Kit, and the nuclei were labeled with DAPI (2 μg/mL) for 10 min. The treated cells were observed using a laser confocal microscope, with the excitation and emission wavelengths of Azide 488 (the fluorescence for labeling EdU) being 495 and 519 nm, respectively.

### 2.10. Intracellular ROS Level Detection

HDF cells were seeded into 24-well plates at a density of 4 × 10^4^ cells/well, with 500 μL per well. After 24 h of culture, the supernatant was discarded. The model control group (MC) was supplemented with DMEM medium containing 0.6 mmol/L H_2_O_2_, while the free SA and SA-NLPs groups received H_2_O_2_ plus test samples (40 μg/mL SA). The NC group without H_2_O_2_ treatment was also set up. After 24 h, cells were incubated with DMEM containing 20 µM 2′,7′-Dichlorofluorescin diacetate (DCFH-DA) for 20 min, washed three times with PBS, and visualized under a fluorescence microscope (DCF excitation/emission: 488/525 nm). The fluorescence intensity was further analyzed using flow cytometry.

### 2.11. Anti-Aging Factors Secretion Level Detection

HDF cells were cultured into 24-well cell culture plates a 4 × 10^4^ cells/well for 24 h and treated with SA-NLPs or free SA at SA concentrations of 20, 40, or 80 μg/mL. The NC and MC groups received 500 μL DMEM only, while the PC group was treated with 40 μg/mL VC. After incubation for 1 h, the cells except NC group were then irradiated with UVA (Philips, Amsterdam, The Netherlands) at 2 × 10^5^ J/m^2^. Following 24 h of culture, the supernatant was collected, and the levels of MMP-1, MMP-3, Col-I, and Col-III were measured using ELISA kits.

### 2.12. Tyrosinase Activity and Melanin Content Detection

B16 cells were seeded into 24-well cell culture plates at 4 × 10^4^ cells/well, cultured for 24 h, and treated with SA-NLPs or free SA (SA concentrations of 20, 40, or 80 μg/mL) in the presence of 100 nmol/L α-MSH for 24 h. NC and MC groups were treated with DMEM or α-MSH alone, respectively, while the PC group received α-MSH + 40 μg/mL VC. After 24 incubation, tyrosinase activity and melanin content were determined using ELISA kits.

### 2.13. Barrier Protection and Moisturizing Factors Detection

HaCaT cells were seeded into 24-well cell culture plates at a density of 4 × 10^4^ cells per well for 24 h. Subsequently, the cells were treated with SA-NLPs or free SA at SA concentrations of 20, 40, or 80 μg/mL. The PC group was treated with 5 mmol/L glycerol, and the NC group received medium only. After 24 h, the cells were harvested, centrifuged, and lysed. The levels of FLG, AQP3, CLDN-1, and HA were quantified using ELISA kits.

### 2.14. 3D Skin Model Barrier Repair, Anti-Aging, and Skin-Brightening Study

The anti-aging effect was evaluated using Ex-vivo^®^, a full-thickness ex vivo human skin model retaining native skin histological architecture (epidermis, dermis, stratum corneum). Following overnight equilibration on nutrient agar in 6-well plates at 37 °C, 5% CO_2_, and ~95% humidity, models were randomized into four groups. The NC group, PC group, free SA group (0.03% SA), and SA-NLPs group (1% SA-NLPs, equivalent to 0.03% SA) were exposed to UV irradiator (Philips) with UVA (320–400 nm, 3 × 10^5^ J/m^2^) and UVB (280–320 nm, 500 J/m^2^), while no treatment was performed in the BC group. For the PC group, Ex-vivo^®^ was treated with 100 μg/mL VC and 7 μg/mL VE after UVA + UVB simulation. After incubation for 24 h, Ex-vivo^®^ was fixed with 4% paraformaldehyde for 24 h, and immunofluorescence of the COL-I and COL-IV was detected by fluorescence microscope (excitation/emission: 488/518 nm).

Commercially available MelaKutis^®^, featuring melanocyte integration and UV-inducible melanogenesis, was utilized to assess skin-brightening effect. After overnight equilibration, the NC group, PC group, free SA group, and SA-NLPs group were stimulated with UVB (500 J/m^2^) for 24 h, while no treatment was carried out in the BC group. For the PC group, MelaKutis^®^ was treated with kojic acid (500 μg/mL) after UVB simulation. After 24 h of incubation, images of MelaKutis^®^ were captured and analyzed, and the L* value and melanin content were quantified using a colorimeter and alkaline lysis method.

EpiKutis^®^ (Guangdong Biocell Biotechnology Co., Ltd., Dongguan, China), a reconstructed human epidermis model histologically analogous to native tissue, was employed for barrier repair assessment. The EpiKutis^®^ was randomly divided into the blank control group (BC), NC group, PC group, free SA group, and SA-NLPs group, with three replicates in each group. Except for the BC group without any treatment, EpiKutis^®^ in the NC group, PC group, and free SA and SA-NLPs groups were stimulated with 0.2% SLS. For the PC group, EpiKutis^®^ was treated with WY-14643 (pirinixic acid) after SLS simulation. After incubation for 24 h, residues on the surface of EpiKutis^®^ were washed with sterile PBS solution, fixed with 4% formaldehyde, embedded in paraffin, and sectioned. The sections were stained with hematoxylin and eosin (H&E) for histological analysis. For immunofluorescence assays, the models were fixed with 4% paraformaldehyde for 24 h, and immunofluorescence of the FLG, LOR, and CLDN1 was detected by fluorescence microscope. The excitation and emission wavelengths were 488 and 518 nm, respectively.

### 2.15. A Zebrafish Model

#### 2.15.1. Antioxidant Effect Evaluation in Zebrafish

Adult healthy zebrafish (Danio rerio, the China Zebrafish Resource Center) were acclimated for two weeks in water maintained at (27 ± 1) °C, pH 6.5–8.5, with room temperature controlled at 20–25 °C. A photoperiod of 12–16 h per day was ensured, alongside a well-functioning filtration system. The fish were fed at least twice daily, with intervals of no less than 3 h between feedings to prevent overfeeding. Oxidative damage was induced in wild-type zebrafish embryos using a hydroxyacetone model. Wild-type zebrafish embryos were pretreated with 0.03 mg/mL phenylthiourea (PTU) to inhibit melanogenesis and maintained until 72 h post-fertilization (hpf). Embryos were distributed into a 24-well plate (10 embryos per well) with three replicates per group. The embryos in the NC group were treated with Holt buffer (a buffer solution for embryos) only. The MC group was treated with 0.025 g/L hydroxyacetone, and the PC group was treated with 0.025 g/L hydroxyacetone and 0.1 g/L glutathione. For the SA-NLPs and free SA groups, embryos were treated with 0.025 g/L hydroxyacetone containing SA-NLPs or free SA with SA concentration of 300 μg/mL, respectively. After 2 h incubation at 28 ± 1 °C in the dark, the treatment solutions were discarded, and the embryos were gently washed three times with Holtfreter’s buffer. After staining with 2 mL of 5 μM H_2_DCFDA (a fluorescent ROS probe) in the dark for 1 h, the embryos were washed three times and incubated with 2 mL of fresh Holtfreter’s buffer for 30 min under dark conditions. Intracellular ROS level was determined by fluorescence using a fluorescence microscope (DCF excitation/emission: 488/525 nm).

#### 2.15.2. Whitening Effect Evaluation in Zebrafish

Zebrafish embryos at 24 hpf were utilized for whitening effect evaluation. A 24-well plate was employed, with 10 embryos per well. Each group was set up with three replicates. The embryos in the NC group were treated with Holt buffer only, and free SA and SA-NLPs groups were treated with free SA or SA-NLPs with a SA concentration of 300 μg/mL, respectively. The PC group was treated with 2.5 g/L kojic acid. Following 48 h incubation, the embryos were embedded with 2–4% methylcellulose with their dorsal sides facing up and imaged under a stereomicroscope.

#### 2.15.3. Analysis of the Impact on Col1a2 and ELNA Genes Related to Anti-Aging in Zebrafish

Zebrafish embryos at 24 hpf were allocated to a 24-well plate with 10 embryos per well. Each group consisted of three replicates. The embryos in the NC group were treated with Holt-Buffer only, and free SA and SA-NLPs groups were treated with free SA or SA-NLPs with a SA concentration of 300 μg/mL, respectively. After incubation for 48 h, the embryos were harvested and washed with Holt buffer to remove residual reagents, and 20 zebrafish from each well were collected. RNA was extracted using the FastPure Cell/Tissue Total RNA Isolation Kit V2, and reverse-transcribed into cDNA using the HiScript III RT SuperMix for qPCR (+gDNA wiper). Real-time PCR amplification was performed using PerfectStart^®^ Green qPCR SuperMix (Takara Bio Inc. Shiga, Japan), with GAPDH serving as the reference gene for relative expression analysis.

### 2.16. Clinical Trial

The clinical research was conducted at Mageline Asia-Pacific R&D center in Wuhan, China. The research protocol was examined and approved by the China-norm Ethics Committee for Clinical Research (No. YWS20250225A). All participants were informed of the potential benefits, risks, and possible complications prior to enrollment, and written informed consent was obtained from each subject. A total of 30 Chinese volunteers, aged between 25 and 60 years and presenting visible wrinkles or fine lines, were enrolled. Participants were randomly assigned to two groups: Group A was treated with a cream containing 1% SA-NLPs (equivalent to 0.03% SA); while Group B received a cream containing free SA at 0.03%. In addition to SA, the cream contained the following components: water, glycerin, isopropyl isostearate, caprylic/capric triglyceride, butylene glycol, butyrospermum parkii (shea) butter, dimethicone, cetearyl alcohol, stearyl alcohol, arachidyl alcohol, and cetearyl glucoside. Both formulations were applied twice daily, in the morning and evening, for a period of 56 days. Assessments were performed at baseline (D0), day 14 (D14), day 28 (D28), and day 56 (D56).

Prior to each evaluation, subjects were required to remain seated calmly in a standardized environment with controlled temperature (20–22 °C) and humidity (40–60%) for at least 30 min. Skin tone parameters (L* and b* values) were analyzed using Image-Pro Plus 6 software with the DP-400 data processor (KONICA MINOLTA, Tokyo, Japan). Transepidermal water loss (TEWL) was measured using a Vapometer (SWL5338, Delfin Technologies Ltd., Kuopio, Finland). Wrinkle-related parameters, including wrinkle length and wrinkle area, were evaluated using the C-Cube system (Pixience, Plano, TX, USA). Standardized facial images were captured using the VISIA imaging system (Canfield Scientific, Parsippany, NJ, USA) to assess overall skin condition.

### 2.17. Statistical Analysis

All results are shown as mean ± SD from at least three independent experiments. Statistical analysis was performed with one-way ANOVA. *p* < 0.05 was considered statistically significant.

## 3. Results

### 3.1. Characterization of SA-NLPs

The prepared SA-NLPs displayed a transparent, pale yellow color, with a particle size of 50.43 nm (Figure 1A), PDI of 0.107, and zeta potential of −36.5 mV. The TEM image in Figure 1B shows that the SA-NLPs exhibited spherical shape with a distinct bilayer structure, with the particle size consistent with the DLS data. Consequently, under these optimized conditions, the SA-NLPs achieved an *EE* as high as 88.2 ± 0.4%, with a *DLE* of 2.65 ± 0.16%. Additionally, a representative measurement report confirming the quality of the data has been provided in the Appendix A.

Low stability remains one of the main limitations that hampers the application of SA. The stability study was carried out at various conditions, including room temperature (RT), 4 °C, −20 °C, 45 °C, and exposure to light. As seen in Figure 1C, free SA without nano-encapsulation changed its color within storage, especially at 45 °C. In comparison, no appearance change was noticed in the SA-NLPs at all the storage conditions for 28 days, indicating excellent stability. The particle size, PDI, and zeta potential of SA-NLPs during storage under various conditions are shown in Appendix A. After 28 days of storage under various conditions, the changes in particle size, PDI, and zeta potential of SA-NLPs were negligible. The relatively stable zeta potential and sufficiently low PDI contributed to the high stability of the bilayer structure, effectively reducing the possibility of self-aggregation during storage and thereby enhancing stability. In summary, the developed SA-NLPs exhibit advantages such as uniform small particle size, high *DLE* and *EE*, and excellent stability, making them suitable for scalable industrial applications in transdermal delivery systems.

### 3.2. Biocompatibility Evaluation

In vitro cytotoxicity profiles of free SA and SA-NLPs at varying concentrations against HaCaT and HDF cells are presented in Figure 2A,B. Over the SA concentration range of 10–160 μg/mL, both free SA and SA-NLPs maintained cell viability above 95% in both cell lines, demonstrating an absence of cytotoxic effects. The biocompatibility of SA-NLPs was further validated through the HET-CAM irritation assay. As shown in Figure 2C and Appendix A, neither free SA nor SA-NLPs induced hemolysis at SA concentrations up to 3 mg/mL after 300 s of exposure. Notably, SA-NLPs consistently exhibited a reaction score of 0.07, indicating minimal irritation and confirming their favorable safety profile.

### 3.3. In Vitro Release and Skin Permeation

The in vitro release kinetics of free SA and SA-NLPs were systematically characterized (Figure 3A). Notably, SA-NLPs demonstrated a substantially attenuated release profile, with only 63.60% of SA released over 12 h compared to the rapid release of 92.25% observed for free SA (*p* < 0.05), confirming the formulation’s sustained-release capacity. This controlled release mechanism is pharmacologically advantageous for skincare applications, as prolonged exposure to therapeutic SA concentrations enhances sustained bioactivity in skin.

A critical limitation of conventional transdermal delivery systems lies in their inadequate cutaneous permeation efficiency. To address this, SA was encapsulated into NLPs to enhance its skin permeation. Franz diffusion cell assays revealed that SA-NLPs significantly improved dermal bioavailability. Quantitative analysis demonstrated a 246.98% increase in skin retention and a 340.10% enhancement in cumulative skin permeation compared to free SA (Figure 3B,C), establishing that NLPs encapsulation notably enhances both skin accumulation and permeation of bioactive compounds.

The skin permeation behavior was further elucidated through confocal microscopy using RhoB as a tracer molecule. While Free RhoB remained confined to the stratum corneum at 2 h post-application, RhoB-NLPs breached the SC barrier within 2 h, achieving a penetration depth of 200.92 μm (Figure 3D). Progressive time-dependent enhancement was observed, with RhoB-NLPs reaching a maximum dermal penetration depth of 430.16 μm at 6 h—statistically superior to Free RhoB (*p* < 0.05) (Appendix A). Depth skin permeation is critical for anti-aging application, as effective anti-aging interventions require bioactive deposition not only in the viable epidermis but specifically within dermal fibroblast populations. These data illuminate that SA-NLPs could overcome key limitations of free drug administration through sustained release, and enhanced skin permeability to reach the dermal area via the advantage of nanoencapsulation. These characteristics of SA-NLPs suggest their potential as a promising platform for skincare.

### 3.4. Cellular Uptake

The cellular uptake behavior was observed using fluorescence microscope with RhoB as a fluorophore. As shown in Figure 4A,B, after 2 h of co-incubation, HDF and B16 cells treated with free RhoB exhibited minimal detectable fluorescence. In contrast, remarkably stronger red fluorescence was noticed in both HDF and B16 cell lines treated with RhoB-NLPs at the equivalent RhoB concentration. The fluorescence intensity increased with the extension of incubation time. After 4 h of treatment, the fluorescence intensity of the RhoB-NLPs group in both cell lines was further enhanced. The cellular uptake was further quantified by flow cytometry. As shown in Figure 4C, the mean fluorescence intensity (MFI) of RhoB-NLPs-treated HDF cells after 4 h incubation represented a 1.80-fold increase over free RhoB controls (*p* < 0.01). Similarly, B16 cells exhibited 2.06-fold greater MFI with RhoB-NLPs versus Free RhoB (*p* < 0.01), confirming NLPs-enhanced uptake across divergent cell types. These results indicate that the encapsulation of NLPs would facilitate the more efficient cellular uptake of the incorporated ingredient by HDF and B16 cells, thus increasing both the uptake efficiency and the intracellular accumulation of the active ingredients within the target skin cells.

### 3.5. Cell Proliferation

The cell proliferation effect of free SA and SA-NLPs was assessed on HDF and HaCaT cells. Figure 5A,B show that at a SA concentration of 40 μg/mL, no remarkable difference in cell activity was noticed between free SA and the NC group, while both cells treated with SA-NLPs exhibited significantly enhanced cell activity compared to the NC group (*p* < 0.05). Once increasing the SA concentration to 80 μg/mL, both formulations demonstrated significant pro-proliferative activity compared to the NC group, with SA-NLPs exhibiting superior bioenhancement compared to free SA (*p* < 0.05). The cell proliferative advantage was mechanistically validated through the EdU assay. As seen in confocal microscopy images (Figure 5C,D), the EdU-positive rate of HDF and HaCaT cells treated with SA-NLPs showed a substantial increase, with 24.13% and 13.78% higher than that treated with free SA (*p* < 0.01) (Appendix A), indicating that the SA-NLPs promoted cell proliferation more effectively than free SA at equivalent dosage. These results indicate that NLPs encapsulation of SA considerably improved its cellular bioavailability.

### 3.6. Antioxidative Effect Evaluation

Overproduction of ROS leads to oxidative stress, causing aging, inflammation, and skin damage [23,24]. The antioxidative effect was investigated on H_2_O_2_-damaged HDF cells. As seen in Figure 6A, after being treated with 0.6 mmol/L H_2_O_2_, strong green fluorescence was observed in the MC group, indicating the successful establishment of the oxidative damage cell model. Following the administration of free SA and SA-NLPs, the intracellular ROS fluorescence intensity was reduced. Figure 6B depicts the MFI of cells treated with free SA and SA-NLPs; both considerably decreased compared to the MC group (*p* < 0.01), suggesting that SA exhibited an antioxidative effect. Compared with free SA, the MFI of cells treated with SA-NLPs notably decreased (*p* < 0.05), demonstrating the superior efficacy of SA-NLPs in protecting cells from oxidative stress-induced damage.

### 3.7. Cellular Efficacy Evaluation

Decreasing MMP activity while promoting collagen synthesis is essential for preserving skin health and mitigating aging-related changes. Overexpression of MMP-1 and MMP-3 accelerates collagen degradation, disrupts the structural integrity of the extracellular matrix, and compromises skin elasticity and repair capacity, and manifests as wrinkles and sagging [25,26]. The cellular aging model was established by treatment with 2 × 10^5^ J/m^2^ UVA. As shown in Figure 7A–D, the expression levels of MMP-1 and MMP-3 were elevated, while those of COL-I and COL-III were reduced in the MC group, indicating the successful establishment of the model. Treatment with free SA and SA-NLPs subsequently reduced the activity of MMP-1 and MMP-3 while increasing the levels of Col-I and Col-III in a dose-dependent manner. This suggests that both free SA and SA-NLPs can inhibit MMP activity and enhance collagen synthesis, exhibiting anti-aging effects.

Melanin plays a critical role in protecting the skin by absorbing UV radiation and neutralizing free radicals. Tyrosinase is the key enzyme in melanin synthesis, catalyzing the initial steps of melanin production. Overproduction of melanin and overactivity of tyrosinase can lead to hyperpigmentation. Thus, reducing tyrosinase activity and melanin production is essential for addressing skin pigmentation [27,28]. To investigate the cellular skin-whitening effect, B16 melanoma cells were stimulated with 100 nmol/L α-MSH for 24 h to induce melanin production. As shown in Figure 7E,F, the tyrosinase activity and melanin content in the MC group significantly increased. The tyrosinase activity and melanin production were effectively decreased by treatment with free SA and SA-NLPs in a dose-dependent manner. Compared to the PC group, both free SA (80 μg/mL SA) and SA-NLPs (80 μg/mL SA) secreted lower melanin content. Specifically, free SA demonstrated a 1.1-fold improvement, while SA-NLPs showed a 1.15-fold improvement over the PC group. Although both groups exhibited similar effects, SA-NLPs performed slightly better than free SA. These findings suggest that both free SA and SA-NLPs could inhibit tyrosinase activity and decrease melanin production over a wide concentration range and thus exhibit a skin-whitening effect.

FLG, AQP3, CLND1, and HA are critical regulators of skin barrier functionality. FLG plays a key role in hydration and barrier integrity. AQP3 facilitates water and glycerol transportation to maintain hydration and elasticity. CLND1 strengthens intercellular tight junctions to preserve barrier function and reduce transepidermal water loss [29,30]. HA, a naturally occurring polyanionic polysaccharide, critically regulates epidermal barrier competence by modulating transepidermal water loss and dermal viscoelastic properties [31]. Therefore, enhancing their expression is fundamental for improving skin barrier function and overall skin health. After 48 h of co-incubation with free SA and SA-NLPs, the levels of FLG (Figure 7G), AQP3 (Figure 7H), CLND1 (Figure 7I), and HA (Figure 7J) in HaCaT cells all showed a dose-dependent increase. This indicates that both free SA and SA-NLPs enhance the barrier function of the stratum corneum, promote tight junctions between cells, and facilitate water transport, thereby improving the integrity and hydration capacity of the skin barrier. Moreover, the results demonstrate that SA-NLPs outperform free SA in promoting barrier proteins and moisturizing factors, highlighting their superior potential in protecting and repairing the skin barrier.

### 3.8. 3D Skin Model Efficacy Study

The 3D skin model represents an advanced biomimetic platform characterized by biochemical, mechanical, and structural fidelity, closely recapitulating native tissue physiology, offering superior biological relevance over conventional 2D cultures [32]. In this study, the anti-aging, skin-whitening, and barrier repair effects of SA-NLPs were systematically evaluated by the 3D skin model. COL-I and collagen IV (COL-IV) ensure the firmness, elasticity, and overall functionality of skin. The decrease in these collagens in the skin is a consequence of visible aging signs like wrinkles and reduced skin strength [33,34]. As shown in Figure 8A, in the NC group that was exposed to UVA and UVB radiation, the fluorescence intensity (indicating COL-I and COL-IV expression level) in the 3D skin model was substantially reduced (*p* < 0.01), indicating the successful establishment of the skin aging model. Following treatment with PC (100 μg/mL VC + 7 μg/mL VE), free SA, and SA-NLPs, the fluorescence intensity all increased to various degrees. According to the quantified results in Appendix A, compared with the NC group, the fluorescence intensity of COL-I and COL-IV in the free SA and SA-NLPs group increased drastically by 318.18% and 427.27%, respectively (*p* < 0.01). Notably, the secretion of COL-I and COL-IV in the 3D skin model treated with SA-NLPs remarkably increased by 34.29% and 20.58%, respectively, compared with the free SA. Moreover, compared with the PC group, the COL-I secretion in the SA-NLPs and free SA group was significantly elevated, showing an increase of approximately 54.10% and 14.75%. Additionally, in the case of COL-IV, the results indicate that SA-NLPs exhibited a modest increase of 5.12% over the PC group. Overall, the 3D aging skin model results demonstrate that SA-NLPs could effectively prevent skin aging by enhancing COL-I and COL-IV secretion.

The skin-whitening effect was assessed by the 3D skin-brightening model. The success model construction was confirmed by the notably darker apparent chromaticity of the NC group after UVB stimulation (Figure 8B). The apparent colorimetric increase was in the order of SA-NLPs group > free SA group > NC group. After treatment with the free SA and SA-NLPs, Appendix A shows that the amount of melanin synthesis was reduced by 6.20% and 12.76% compared to the NC group (*p* < 0.05), respectively. In addition, the L* value was increased by 10.73% and 15.84% in the free SA and SA-NLPs groups, respectively, compared with the NC group. Furthermore, when compared with the PC group, SA-NLPs exhibited a 4.49% reduction in melanin content, whereas free SA showed 12.34% higher melanin levels than PC, suggesting a relatively inferior efficacy in melanin suppression. The L* value of the free SA group was 8.74% lower than that of the PC group. In contrast, SA-NLPs demonstrated a 4.54% increase in L* value compared to the PC group. These results suggest that both free SA and SA-NLPs effectively modulate melanin synthesis and skin brightness, with SA-NLPs demonstrating a relatively better effect compared to free SA. (*p* < 0.05) (Appendix A). These results signify that SA-NLPs inhibit tyrosine activity and melanin production in the 3D skin model and thus demonstrate their capability to improve the skin’s apparent luminance and brightness.

FLG, loricrin (LOR), and CLDN1 are indispensable components of the skin barrier, maintaining the skin’s integrity and functionality [35,36]. As presented in Figure 7C, the H&E staining results show that the number of live epidermal cell layers was significantly reduced and the stratum corneum was loosened in the NC group (treated with SLS) compared with the BC group (without any treatment), indicating the break of the skin barrier after SLS treatment. The treatment of free SA and SA-NLPs both improved the SLS-induced barrier damage status, with a superior barrier repair effect in the SA-NLPs group. In addition, the immunofluorescence results reveal that stronger green fluorescence from FLG, LOR, and CLDN1 was detected in the 3D skin treated with SA-NLPs and free SA group compared to the NC group, demonstrating that the 3D skin treated with free SA and SA-NLPs synthesized more FLG, LOR, and CLDN1 to protect the skin. The ensembles of FLG (Appendix A), LOR (Appendix A), and CLDN1 (Appendix A) in the SA-NLPs group increased by 9.40%, 17.07%, and 33.87%, respectively, compared with the free SA group (*p* < 0.01). Notably, compared with the PC group (treated with 2.5 g/L kojic acid), the ensembles of FLG, LOR, and CLDN1 in the SA-NLPs group increased by 6.89%, 7.86%, and 62.74%, respectively. These results validated that SA-NLPs exhibited excellent skin barrier repair functionality by enhancing FLG, LOR, and CLDN1 synthesis. All the 3D skin model results are well-correspondent with previous cellular efficacy results (Figure 7), elucidating the 3D skin model as a promising in vitro research tool.

### 3.9. Efficacy Evaluation in Zebrafish Model

In vivo efficacy was assessed in the zebrafish model. As shown in Figure 9A, the ROS green fluorescence in the MC group (incubated with hydroxyacetophenone) increased compared with the NC group (without treatment), indicating the establishment of the oxidative stress-damaged zebrafish model. Treatment with free SA and SA-NLPs led to decreased fluorescence. According to the quantitative fluorescence analysis (Figure 9B), compared to the MC group, free SA reduced ROS relative fluorescence intensity (RFI) by 24.06%, while the SA-NLPs achieved a higher inhibition rate of 44.16%. Both free SA and SA-NLPs significantly suppressed ROS levels compared to the MC group in the zebrafish model (*p* < 0.01), while SA-NLPs demonstrated a remarkably greater inhibitory effect on ROS levels compared to free SA (*p* < 0.01). These results indicate that SA-NLPs effectively mitigated the damage caused by hydroxyacetophenone-induced oxidative stress and exhibited superior protective effects compared to free SA at equivalent doses.

The skin whitening efficacy was further assessed in the zebrafish model. During zebrafish development, melanocytes begin to grow from the retinal epithelium at 24 h post-fertilization (hpf), forming stable and visible melanin pigment spots by 48 hpf [37]. As shown in Figure 9C, melanin was distributed throughout the whole body of zebrafish under normal culture conditions. Treatment with free SA showed no significant inhibitory effect on melanin synthesis. However, a significant reduction in melanin level was observed in the zebrafish incubated with SA-NLPs compared to the NC group and free SA group (*p* < 0.01), achieving an inhibition rate of 20.17% and 20.16%, respectively (Figure 9D). These results indicate that SA-NLPs effectively suppressed melanogenesis and exhibited superior whitening efficacy compared to free SA at the same dose.

Col1a2 encodes the pro-alpha2 chain of type I collagen, playing an essential role in maintaining skin integrity, facilitating wound healing, and counteracting the degradation of the matrix associated with aging [38,39]. ELNA encodes elastin, is crucial to endow elasticity and resilience of the skin, and contributes to mechanical strength and maintains the barrier function of the skin [40]. Together, Col1a2 and ELNA are pivotal for preserving the structural and functional properties of the skin. Therefore, it is important to evaluate the mRNA expression level of these proteins for anti-aging and skin repair efficacy study [41]. In zebrafish treated with free SA and SA-NLPs, the expression levels of Col1a2 and ELNA were notably upregulated compared to the NC group (*p* < 0.05), with increases of 37.76%, 76.6%, 26.46%, and 57.3%, respectively. Furthermore, SA-NLPs demonstrated a remarkably enhanced promotion rate on mRNA expression levels of both proteins compared to free SA (*p* < 0.05). These findings demonstrate that both free SA and SA-NLPs exhibited anti-aging and skin barrier repair properties, with SA-NLPs showing superior efficacy compared to free SA.

### 3.10. Clinical Efficacy Evaluation

The efficacy of SA-NLPs was further validated in clinical trials. L*, b*, and ITA° are commonly used parameters that objectively measure skin brightness, yellowness, and overall tone to assess whitening effects. As shown in Figure 10A, the group treated with the cream containing SA-NLPs (Group A) exhibited a 4.20% increase in L* and a 7.87% decrease in b* at day 56 (*p* < 0.05). In comparison, Group B, which received the cream containing free SA, showed only a 2.34% increase in L* and a 5.70% reduction in b*. Notably, Group A showed a 27.98% increase in ITA° at day 28 (*p* < 0.005), consistent with the trend observed in L* values, suggesting enhanced skin brightness. In contrast, no significant change in ITA° was observed in Group B (*p* > 0.05). These results indicate that the cream containing SA-NLPs (Group A) provided superior whitening efficacy compared to the cream containing free SA.

TEWL is a widely used indicator of skin barrier function, reflecting the amount of water that passively evaporates through the skin. For skin barrier function, Group A showed a sustained reduction in TEWL, reaching an 8.45% decrease at day 56, with significant interim improvements of 5.19% at day 14 and 7.46% at day 28 (*p* < 0.05) (Table 1). In contrast, Group B exhibited only a 4.75% reduction in TEWL, which was not statistically significant. Moreover, representative VISIA images captured in red mode are shown in Figure 10A, illustrating that Group A was more effective in reducing facial redness compared to Group B, indicating superior skin barrier improvement.

Wrinkles are a primary clinical indicator of skin aging. Therefore, quantitative assessment of wrinkle parameters is essential for objectively evaluating anti-aging efficacy. In this study, Group A, treated with the SA-NLPs cream, demonstrated significantly greater reductions in wrinkle length and area compared to Group B (Figure 10B and Table 1). Remarkably, Group A exhibited a 4.01% reduction in wrinkle length at day 56 (*p* < 0.05), whereas Group B showed a non-significant change of only 0.18% (*p* > 0.1). Similarly, the wrinkle area in Group A decreased by 2.11% at day 56, compared to a 0.72% reduction in Group B, which was also not statistically significant (*p* > 0.1). The anti-wrinkle effect of Group A was approximately 2.93 times greater than that of Group B. These findings collectively demonstrate the superior efficacy of SA-NLPs in reducing wrinkles and further support their advantages in terms of skin brightening and barrier function restoration over free SA.

## 4. Conclusions

In this research, SA-NLPs were developed as a transdermal delivery system to improve the transdermal permeability of SA and enhance its anti-aging, skin-brightening, and barrier repair properties. The fabricated SA-NLPs featured a small, uniformly distributed nanoparticle size, excellent stability, high drug loading, and encapsulation efficiency. Moreover, these particles exhibited enhanced transdermal penetration capabilities and increased cellular uptake. A comprehensive investigation was conducted across four distinct experimental levels: the cellular level, the 3D skin model, the zebrafish study, and the clinical trial. This multi-level approach not only elucidated the efficacy and underlying mechanisms of SA in anti-aging, skin-brightening, and barrier repair processes but also confirmed that SA-NLPs significantly boost the bioavailability of SA by facilitating more efficient skin permeation and promoting greater uptake by relevant target cells. In summary, the developed SA-NLPs have proven to be an effective transdermal delivery solution, significantly enhancing the anti-aging, skin-brightening, and barrier repair effects of SA. Thus, this study provides guidance for their practical application.

## Figures and Tables

**Figure 1 pharmaceutics-17-00956-f001:**
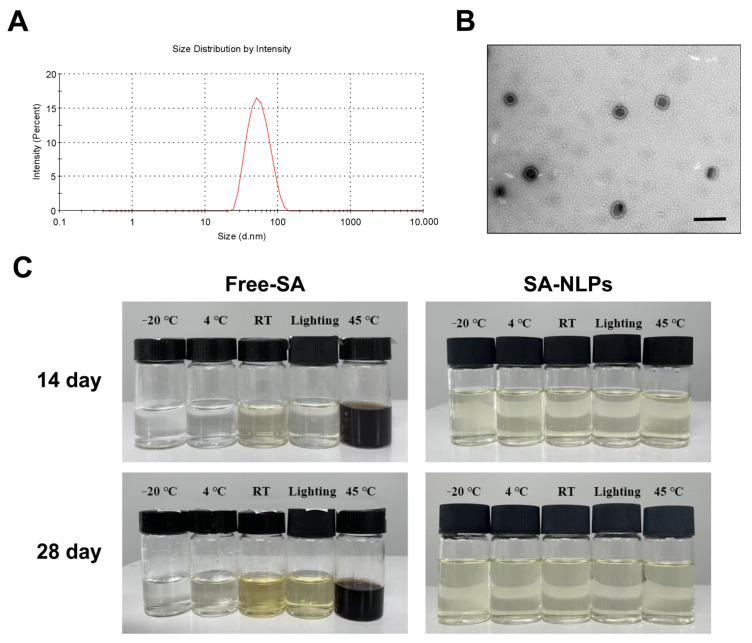
Characterization of SA-NLPs: (**A**) the particle size distribution of the SA-NLPs measured by DLS; (**B**) TEM image of the SA-NLPs (scale bar: 100 nm); (**C**) appearance of free SA and SA-NLPs after storage for 14 and 28 days.

**Figure 2 pharmaceutics-17-00956-f002:**
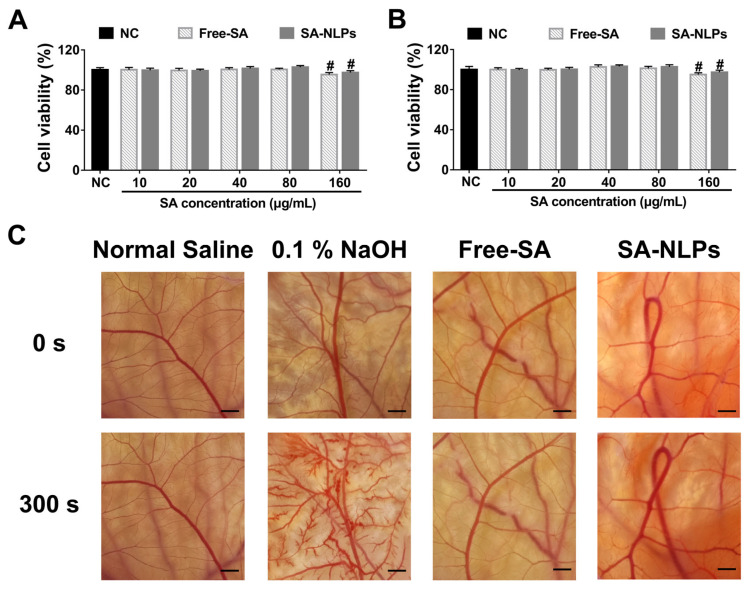
In vitro cytotoxicity of the free SA and SA-NLPs against (**A**) HDF and (**B**) HaCaT cells detected by CCK-8. The cells were treated with the free SA or SA-NLPs at the same SA concentrations over a wide range (10~160 µg/mL). ^#^
*p* < 0.05, vs. NC. Mean ± SD, *n* = 5. (**C**) Evaluation of the irritation potential of free SA and SA-NLPs using the HET-CAM assay. Fertilized hen eggs were treated with free SA or SA-NLPs at a concentration of 10% (equivalent to SA at 3 mg/mL) to assess vascular responses indicative of irritation. Scale bar: 2 mm.

**Figure 3 pharmaceutics-17-00956-f003:**
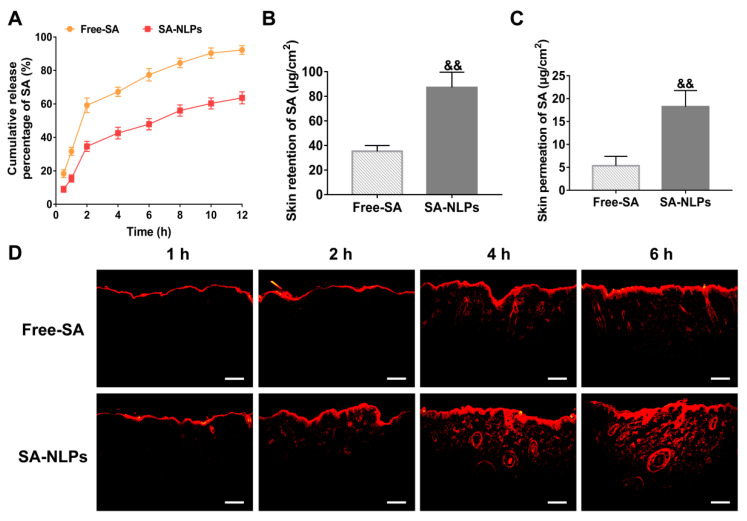
(**A**) In vitro release profiles of SA from free SA and SA-NLPs for 12 h. Evaluation of transdermal permeability of free SA and SA-NLPs. (**B**) In vitro skin retention and (**C**) cumulative permeation of SA. ^&&^
*p* < 0.01 vs. Free. Mean ± SD, *n* = 3. (**D**) Fluorescence microscope images showing RhoB delivery pathway in porcine skin during 6 h of in vitro permeation. Scale bar: 100 μm.

**Figure 4 pharmaceutics-17-00956-f004:**
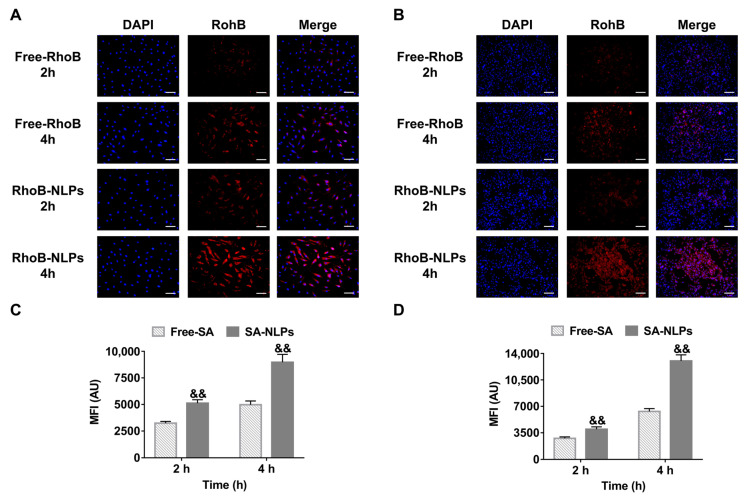
Cellular uptake of free SA and SA-NLPs. Visualization of cellular uptake of RhoB by (**A**) HDF and (**B**) B16 using a fluorescence microscope after incubation for 2 and 4 h. Scale bar: 100 μm. Flow cytometry analysis of MFI in (**C**) HDF and (**D**) B16 cells treated with free SA and SA-NLPs. ^&&^
*p* < 0.01 vs. free SA. Mean ± SD, *n* = 3.

**Figure 5 pharmaceutics-17-00956-f005:**
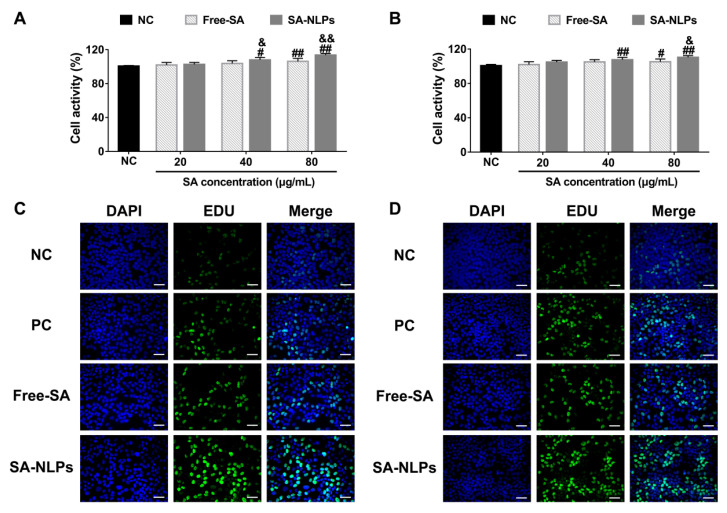
Effect of free SA and SA-NLPs on cell proliferation. Cell viability of (**A**) HDF and (**B**) HaCaT cells treated with different formulations at a wide range of free SA and SA-NLPs for 48 h. ^#^
*p* < 0.05, ^##^
*p* < 0.01 vs. NC, ^&^
*p* < 0.05, ^&&^
*p* < 0.01 vs. Free. Mean ± SD, *n* = 5. Visualization of EdU-processed (**C**) HDF and (**D**) HaCaT using a fluorescence microscope. Scale bar: 100 μm.

**Figure 6 pharmaceutics-17-00956-f006:**
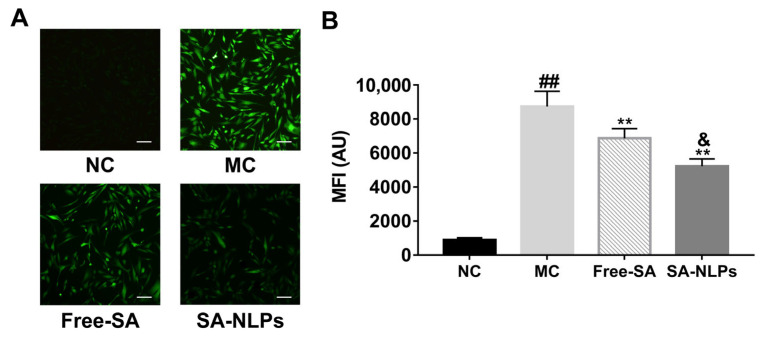
(**A**) Fluorescence microscope images of HDF cells following oxidative stress induction with 0.6 mmol/L H_2_O_2_, subsequently treated with free SA and SA-NLPs. Scale bar: 100 μm. (**B**) The HDF cells were further analyzed by flow cytometry and presented as relative MFI. ^##^
*p* < 0.01 vs. NC; ** *p* < 0.01 vs. MC. ^&^
*p* < 0.05, vs. Free. Mean ± SD, *n* = 3.

**Figure 7 pharmaceutics-17-00956-f007:**
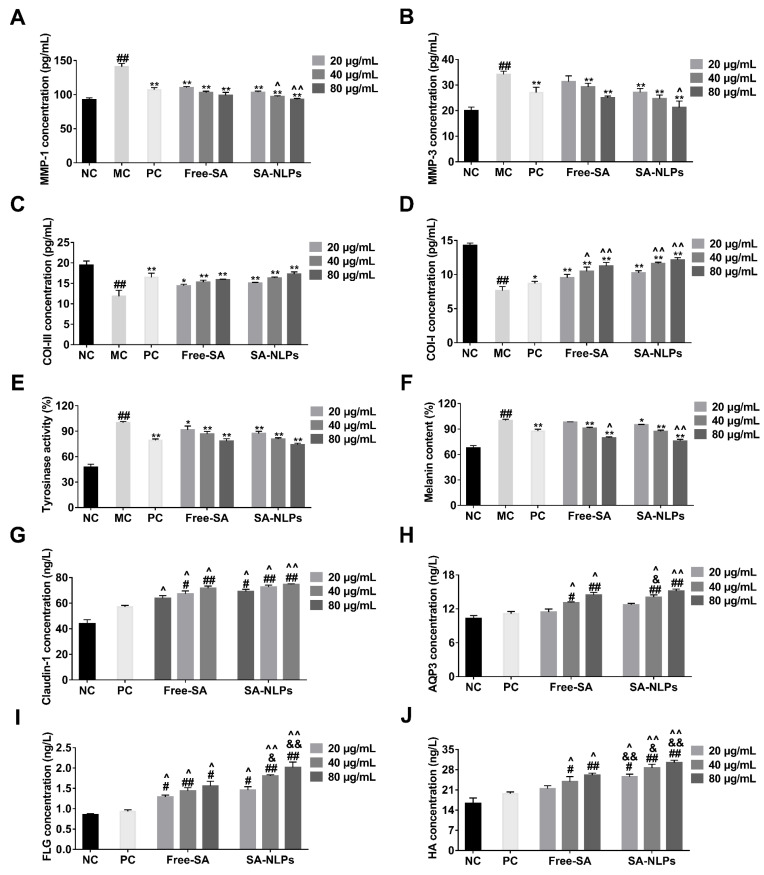
The content of (**A**) MMP-1, (**B**) MMP-3, (**C**) COL-III, and (**D**) COL-I in HDF, which was induced to aging with UVA at 2 × 10^5^ J/m^2^, subsequently treated with free SA and SA-NLPs. The content of (**E**) tyrosinase and (**F**) melanin in B16, which was induced to produce melanin with 100 nmol/L α-MSH, subsequently treated with free SA and SA-NLPs. The content of (**G**) claudin-1, (**H**) AQP3, (**I**) FLG, and (**J**) HA in HaCaT, which was treated with free SA and SA-NLPs. ^#^
*p* < 0.05, ^##^
*p* < 0.01 vs. NC; * *p* < 0.05, ** *p* < 0.01 vs. MC; ^^^
*p* < 0.05, ^^^^
*p* < 0.01 vs. PC; ^&^
*p* < 0.05, ^&&^
*p* < 0.01 vs. Free. Mean ± SD, *n* = 3.

**Figure 8 pharmaceutics-17-00956-f008:**
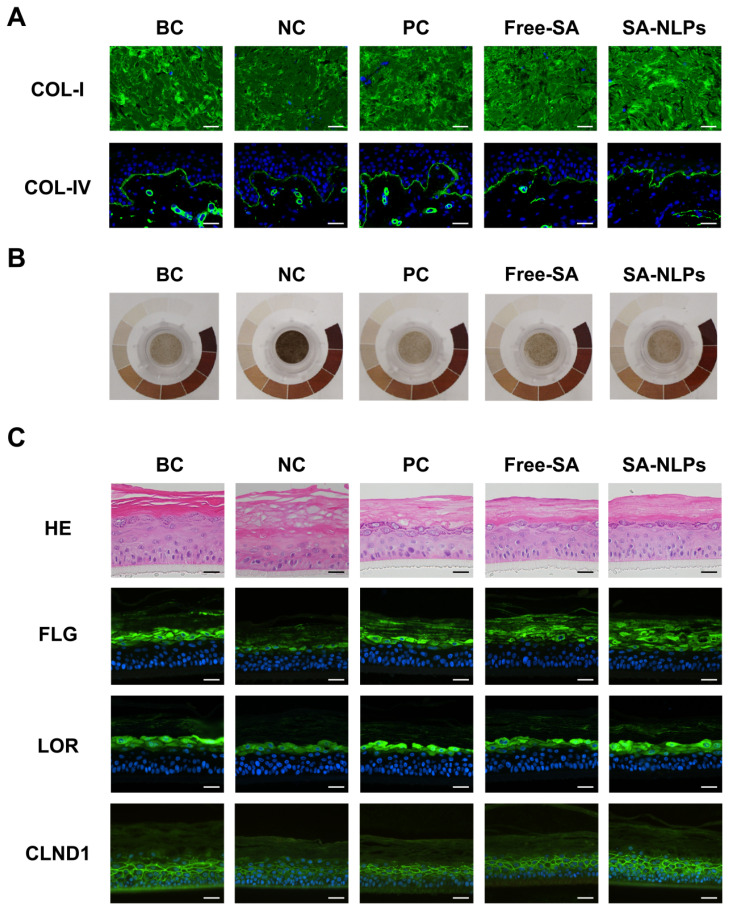
The barrier repair and anti-aging effect of SA studied using the 3D skin models. (**A**) The relative value of synthesized COL-I and COL-IV in Ex-vivo^®^ observed by immunofluorescence microscopy. Scale bar: 50 μm. Except for the BC group (without treatment), the other groups were all subjected to continuous irradiation of UVA (3 × 10^5^ J/m^2^) + UVB (500 J/m^2^) to induce aging. (**B**) Fluorescence microscopy images of the MelaKutis^®^ model after UVB exposure at 500 J/m^2^ for 24 h, followed by treatment with free SA or SA-NLPs at the same SA concentration. (**C**) The HE staining and fluorescent immunization images of FLG, LOR, and CLDN1 in the EpiKutis^®^ model treated with 0.2% SLS, free SA, or SA-NLPs with the same SA concentration, respectively. Scale bar: 50 μm.

**Figure 9 pharmaceutics-17-00956-f009:**
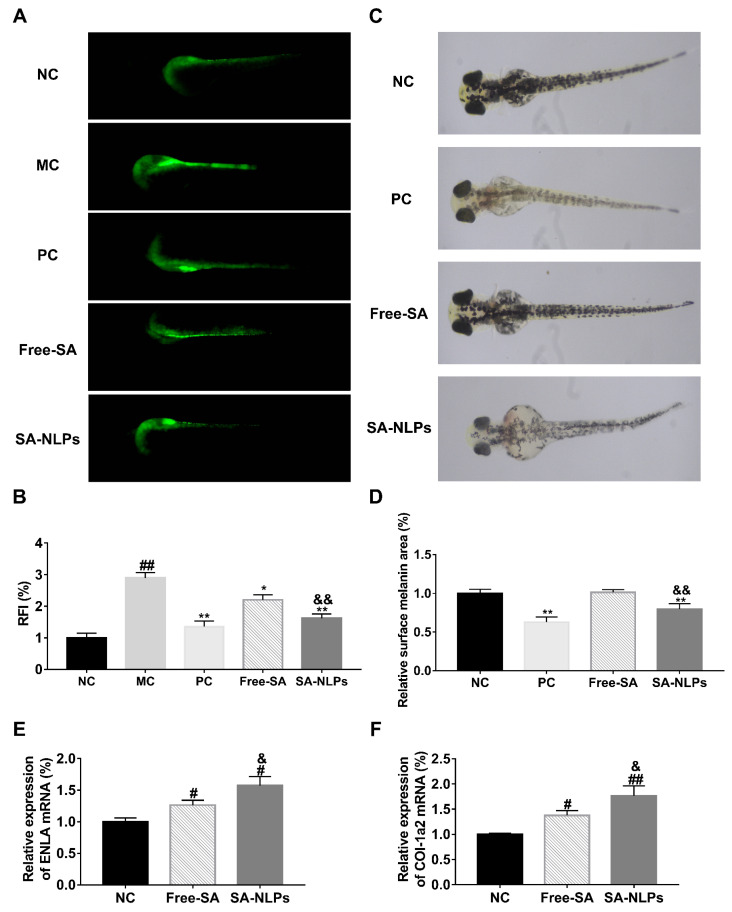
In vivo efficacy evaluation in the zebrafish model: (**A**) fluorescence images of zebrafish embryos after H_2_O_2_ induction, followed by treatment with free SA and SA-NLPs; (**B**) the fluorescence intensity showing ROS levels, quantified and presented using RFI; (**C**) optical microscopy images of melanin distribution; (**D**) relative surface melanin area in zebrafish embryos after intervention with free SA and SA-NLPs; relative expression levels of (**E**) ENLA and (**F**) COL-1a2 gene after treatment with free SA and SA-NLPs. ^#^
*p* < 0.05, ^##^
*p* < 0.01 vs. NC; * *p* < 0.05, ** *p* < 0.01 vs. MC; ^&^
*p* < 0.05, ^&&^
*p* < 0.01 vs. Free. Mean ± SD, *n* = 3.

**Figure 10 pharmaceutics-17-00956-f010:**
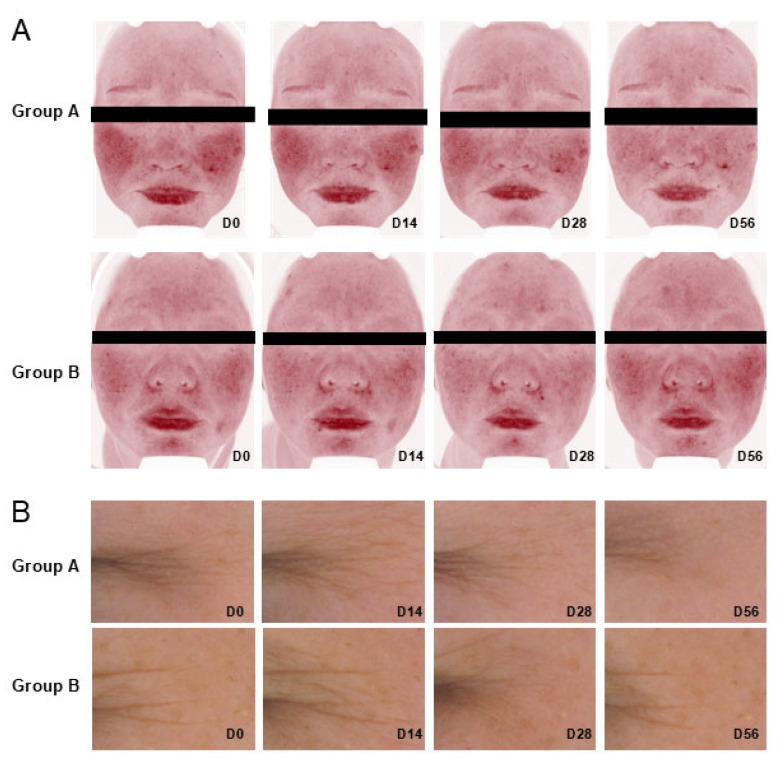
Clinical efficacy of creams containing SA-NLPs: (**A**) representative VISIA images demonstrating improvements in skin redness at days 0, 14, 28, and 56; (**B**) changes in facial wrinkles in Group A and Group B at days 0, 14, 28, and 56, as assessed by C-Cube analysis.

**Table 1 pharmaceutics-17-00956-t001:** Changes in skin parameters in Group A and Group B after cream application at different time intervals.

Skin Parameters	Group	Average Value				Change Rate		
		D0	D14	D28	D56	△D14	△D28	△D56
L*	Group A	47.34 ± 5.51	48.54 ± 5.25	49.11 ± 3.42	49.33 ± 4.33	+2.53%	+3.72%	+4.20%
	Group B	48.27 ± 5.97	46.91 ± 5.23	47.15 ± 6.03	47.14 ± 5.86	−2.82%	−2.31%	−2.34%
b*	Group A	24.0 ± 3.14	22.40 ± 2.85	22.29 ± 2.57	22.18 ± 2.72	−6.95%	−7.43%	−7.87%
	Group B	23.67 ± 4.03	22.89 ± 3.53	22.58 ± 3.52	22.32 ± 2.05	−3.33%	−4.61%	−5.70%
ITA○	Group A	9.28 ± 8.65	10.62 ± 8.50	11.88 ± 8.39	10.72 ± 8.55	+14.45%	+27.98%	+15.50%
	Group B	10.62 ± 8.52	14.00 ± 6.98	10.44 ± 8.46	10.42 ± 8.47	+29.77%	−1.68%	−1.85%
TEWL	Group A	20.30 ± 2.62	19.25 ± 2.30	18.78 ± 2.63	18.58 ± 2.62	−5.19%	−7.46%	−8.45%
	Group B	20.67 ± 4.48	20.15 ± 3.69	19.98 ± 2.38	19.68 ± 2.47	−2.52%	−3.31%	−4.75%
Wrinkle length	Group A	14.81 ± 0.19	14.79 ± 0.40	14.66 ± 0.48	14.22 ± 0.65	−0.17%	−1.01%	−4.01%
	Group B	14.85 ± 0.19	14.92 ± 0.11	14.83 ± 0.26	14.80 ± 0.29	+0.17%	−0.13%	−0.18%
Wrinkle area	Group A	167.0 ± 3.17	166.34 ± 3.26	165.8 ± 2.63	165.04 ± 2.65	−0.57%	−1.03%	−2.11%
	Group B	167.6 ± 1.95	167.45 ± 1.80	167.1 ± 1.60	166.41 ± 2.52	−0.23%	−0.41%	−0.72%

## Data Availability

Data will be made available on request.

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
