# Peer review of "Sialic Acid-Loaded Nanoliposomes with Enhanced Stability and Transdermal Delivery for Synergistic Anti-Aging, Skin Brightening, and Barrier Repair"

_pharmaceutics, 2025, doi:10.3390/pharmaceutics17080956_

Round 1
Reviewer 1 Report
Comments and Suggestions for Authors
- Reduce the plagiarism up to below 15%
- More quantitative data should be added to the abstract
- The authors should give more details on how they purified the synthesized Sialic Acid Loaded Nanoliposomes.
- Include error bars in all figures
- Cite comparative studies using other nanocarriers for similar cosmeceutical delivery.
- Include more recent references and discuss
- Check grammatical errors throughout the manuscript
- Include the hemolytic assay for the synthesized Sialic Acid Loaded Nanoliposomes.
- Include more TEM images for Nanoliposomes
- Include scale bars for all images in Figure 2C
- Check SI units throughout the manuscript
Comments on the Quality of English Language
Need to be improved
Reviewer 2 Report
Comments and Suggestions for Authors
This paper is a comprehensive study of liposome encapsulation of sialic acid. The breadth of experiments, from stability to induction of skin barrier to antioxidant activity to melanogenesis inhibition is quite striking. In order to make this paper more convincing, details must be included that are lacking:
L140: what is TGC?
L143: what is the ratio of addition of the phases?
L147: what is the pore size of the filter used? This is a critical step in separating encapsulated SA
L163: Where is Wf?
L192, L205: What is the SA concentration?
L280, L 303: What is the light source of the UVA and UVB?
L302, 309, L382, L708: What is the concentration of SA in the formula/cream and what are the other components of the cream?
Fig 3A: No explanation for what assay is "in vitro release"
The activity of SA in these studies does not match the activity reported by Ref 2. Ref 2 reports an EC(50) for B16 tyrosinase inhibition of 7.22 mg/ml, while this paper reports inhibition at 80 mcg/ml, about 100-times less. Ref 2 reports EC(50) antioxidant activity at 1 mg/ml but this paper reports activity at 40 mcg/ml, about 25-times less. How is this possible?
The encapsulation efficiency of 88% achieved by high pressure homogenization of a hydrophilic SA into hydrophobic lecithin is hard to imagine, based on the liposome literature. This is of course dependent on the method of separating the encapsulated SA from the free SA by filtration/centrifugation, which the authors do no describe or give the filter pore size. The authors should explain why they think they achieved such an incredibly high encapsulation of a sugar into a phospholipid mix.
Serum contains SA (Varki, Trends Mol Med 2008 14:351) and this study uses serum in the cell and tissue culturing. The authors should measure the SA in their serum and comment on why they observed such SA effects over and above the SA in serum.
Reviewer 3 Report
Comments and Suggestions for Authors
09.06.2025
A review to evaluate its suitability for publication Type of manuscript:
Article
Title: Sialic Acid Loaded Nanoliposomes with Enhanced Stability and Transdermal Delivery for Synergistic Anti-Aging, Skin Brightening and Barrier Repair
Authors: Fan Yang , Hua Wang , Dan Luo , Jun Deng , Yawen Hu , Zhi Liu * , Wei Liu *
The manuscript Sialic Acid Loaded Nanoliposomes with Enhanced Stability and Transdermal Delivery for Synergistic Anti-Aging, Skin Brightening and Barrier Repair is devoted to the development of a new liposomal dosage form containing sialic acid as the main active ingredient. The authors describe the method for obtaining and stabilising liposomal sialic acid and the in vivo study of the effectiveness of the new dosage form.
In the introduction, the authors describe the characteristics of sialic acid in terms of its molecular structure, hydrophilic-hydrophobic properties, solubility, and biological significance in both neurobiology and cosmetology.
In the introduction, the authors describe the characteristics of sialic acid in terms of its molecular structure, hydrophilic-hydrophobic properties, solubility, and biological significance in both neurobiology and cosmetology.
- Comment/suggestion for improving the quality of the material: Based on the relevance of the Introduction section, the aim of the current study should be formulated.
The Materials and Methods section describes in detail all the approaches used by the authors to carry out this study.
- Comment/suggestion for improving the quality of the material: it is necessary and desirable to present the sialic acid (N-acetylneuraminic acid) molecule based on accepted pharmaceutical standards: Name (International Nonproprietary Names (INN), trade name, etc.); structural formula + chemical name; active ingredient content in the medicinal substance and its degree of purity; expiration date, storage conditions.
- Comment/suggestion for improving the quality of the material: Is sialic acid included in the collection of standards - world pharmacopoeias? What is the current status of this compound? If this substance is not included in the pharmacopoeias standart, is there an official possibility to offer dosage forms based on it for use in medical cosmetology?
- Comment/suggestion for improving the quality of the material: Lines 400-401: When measuring the dynamic properties of colloidal solutions using light scattering (DLS), the dilution of the sample is important in order to avoid obtaining a poor-quality report. How did the authors dilute the initial liposomal system for DLS measurement?
- Comment/suggestion for improving the quality of the material: Tables S1 and S2 should present the results for all three measured parameters, as
- Comment/suggestion for improving the quality of the material: Lines 436, 437: Figure 2 (C) shows the results of processing fertilised eggs with the samples under investigation. The results are interesting and significant for the work, but there is no description of this methodology in the Materials and Methods section.
- Is this methodology a standard procedure? What are the statistics for the results of this in vitro cytotoxicity study?
Overall, the authors' research work represents a complete, logically sound study with clearly defined objectives and results that are beyond doubt in terms of their reliability.
I believe that the work can be considered for publication in Pharmaceutics after the authors have made corrections in accordance with the reviewer's comments.
Respectfully, reviewer
